# Amphiphilic PTB7-Based Rod-Coil Block Copolymer for Water-Processable Nanoparticles as an Active Layer for Sustainable Organic Photovoltaic: A Case Study

**DOI:** 10.3390/polym14081588

**Published:** 2022-04-13

**Authors:** Marianna Diterlizzi, Anna Maria Ferretti, Guido Scavia, Roberto Sorrentino, Silvia Luzzati, Antonella Caterina Boccia, Andrea A. Scamporrino, Riccardo Po’, Eleonora Quadrivi, Stefania Zappia, Silvia Destri

**Affiliations:** 1Istituto di Scienze e Tecnologie Chimiche “Giulio Natta” (SCITEC)—CNR, Sede Via A. Corti 12, 20133 Milano, Italy; marianna.diterlizzi@scitec.cnr.it (M.D.); guido.scavia@scitec.cnr.it (G.S.); r.sorrentino988@gmail.com (R.S.); silvia.luzzati@scitec.cnr.it (S.L.); antonella.boccia@scitec.cnr.it (A.C.B.); 2Dipartimento di Scienza dei Materiali, Università degli Studi di Milano-Bicocca, Via Cozzi 55, 20125 Milano, Italy; 3Istituto di Scienze e Tecnologie Chimiche “Giulio Natta” (SCITEC)—CNR, Sezione Via G. Fantoli 16/15, 20138 Milano, Italy; anna.ferretti@scitec.cnr.it; 4Istituto per i Polimeri, Compositi e Biomateriali (IPCB) U.O.S. di Catania—CNR, Via Gaifami 18, 95126 Catania, Italy; andreaantonio.scamporrino@cnr.it; 5Eni SpA—Renewables, New Energies and Material Science Research Center, “Istituto Guido Donegani”, Via Fauser 4, 28100 Novara, Italy; riccardo.po@eni.com (R.P.); eleonora.quadrivi@eni.com (E.Q.)

**Keywords:** PTB7, rod-coil block copolymer, water-processable nanoparticles, miniemulsion, Janus nanoparticles, NP-OPV

## Abstract

We synthetized a new rod-coil block copolymer (BCP) based on the semiconducting polymerpoly({4,8-bis[(2-ethylhexyl)oxy]benzo[1,2-*b*:4,5-*b′*]dithiophene-2,6-diyl}{3-fluoro-2-[(2-ethylhexyl)carbonyl]thieno[3,4-*b*]thiophenediyl}) (PTB7) and poly-4-vinylpyridine (P4VP), tailored to produce water-processable nanoparticles (WPNPs) in blend with phenyl-C71-butyric acid methyl ester (PC_71_BM). The copolymer PTB7-*b*-P4VP was completely characterized by means of two-dimensional nuclear magnetic resonance (2D-NMR), matrix-assisted laser desorption/ionization-time of flight (MALDI-TOF) mass spectrometry (MS), size-exclusion chromatography (SEC), and differential scanning calorimetry (DSC) to confirm the molecular structure. The WPNPs were prepared through an adapted miniemulsion approach without any surfactants. Transmission electron microscopy (TEM) images reveal the nano-segregation of two active materials inside the WPNPs. The nanostructures appear spherical with a Janus-like inner morphology. PTB7 segregated to one side of the nanoparticle, while PC_71_BM segregated to the other side. This morphology was consistent with the value of the surface energy obtained for the two active materials PTB7-*b*-P4VP and PC_71_BM. The WPNPs obtained were deposited as an active layer of organic solar cells (OSCs). The films obtained were characterized by UV-Visible Spectroscopy (UV-vis), atomic force microscopy (AFM), and grazing incidence X-ray diffraction (GIXRD). J-V characteristics of the WPNP-based devices were measured by obtaining a power conversion efficiency of 0.85%. Noticeably, the efficiency of the WPNP-based devices was higher than that achieved for the devices fabricated with the PTB7-based BCP dissolved in chlorinated organic solvent.

## 1. Introduction

Organic photovoltaics (OPVs) have attracted increasing interest over the last few decades as an alternative to inorganic photovoltaics because they offer a wide range of promising features and have useful electronic properties, including component versatility, as well as low production and installation costs. They enable the production of light-weight, solution-processable flexible devices which can be applied within large area [1]. Many efforts were made by researchers to overcome some of the drawbacks which limit OPV industrial implementation, such as relatively low power conversion efficiencies (PCEs) and long-term stability [2]. Nevertheless, their commercialization is still restricted by the use of huge amounts of hazardous solvents on laboratory scale. Particularly, halogenated organic solvents are toxic and harmful for environment and human health, even though they help to control the morphology during the device fabrication, as well as optimize the interpenetrating network between electron donor and acceptor materials in the active layer [3,4]. For this reason, the scientific community focus on the reduction and/or substitution of these solvents with more sustainable alternatives [5,6,7].

Over the past two decades, different approaches to deposit the active layer with water were developed. Since most semiconducting materials are insoluble in aqueous medium, the production of water-processable nanoparticle (WPNP) dispersions was investigated [8,9]. One of the main advantages related to the WPNP dispersion approach is the creation of nanodomains (~10–20 nm) of donor and acceptor materials compatible with the exciton diffusion length [10,11,12,13,14]. Nanoparticles (NPs) composed of a blend containing organic semiconductor materials and acceptor molecules can be obtained mainly by two approaches: the miniemulsion method and the nanoprecipitation technique. The miniemulsion method exploits the immiscibility of the polymer solvents with the nonsolvent (aqueous phase) to form stable WPNP dispersions [15]. This method involves the use of surfactant molecules to stabilize the droplets from aggregation. The surfactants have electric insulating behavior, and additional steps to remove the excess are required to avoid a drastic drop in the device performance, thus increasing the whole process cost and duration [16]. The WPNPs prepared with this method mainly have core-shell morphology as a consequence of the duration of solvent evaporation [17,18]. Moreover, fullerene materials display surface energy higher with respect to the electron donor polymers which causes the formation of core-shell WPNPs holding fullerene-rich core too [19,20]. Thus, a mild thermal annealing (close to glass transition temperature of the electron donor polymer) is necessary to obtain a highly intermixed donor–acceptor network to enable charge transport and extraction in the active layer.

The nanoprecipitation method involves the rapid injection of active materials dissolved in an organic solvent into an alcoholic medium with the formation of surfactant-free WPNPs with homogeneous distribution of donor and acceptor materials into the nanostructures [21,22]. The absence of the surfactants led to stability issues of the inks because of the WPNP aggregation [8]. Recently, Xie et al. proposed the use of a surfactant-assisted nanoprecipitation approach for the synthesis of WPNPs using a poloxamer as the surfactant. It stabilizes the aqueous dispersion, and the excess can be easily stripped away, overcoming the limits of both the miniemulsion and the nanoprecipitation approaches. The nanoparticle-based OPV (NP-OPV) devices obtained displayed performances and stability comparable to those of devices processed from chlorinated organic solvents and NP-based organic photovoltaics (NP-OPVs) which achieved an efficiency of 7.5% [23].

An alternative way to produce WPNP dispersions is based on the exploitation of amphiphilic rod-coil block copolymers (BCPs). BCPs emerged as a powerful tool to achieve ideal morphologies in OPV devices because of their self-assembly ability [24,25,26]. In particular, the amphiphilic rod-coil BCPs are materials constituted by a conjugated rigid block covalently linked to a hydrophilic flexible segment. Hence, they are able to self-assemble in aqueous medium without any surfactant on the basis of the physical-chemical behavior to form phase separation at the nanoscale without segregation at the macroscopic level whilst minimizing unfavorable interactions [27]. Recently, our research group reported on amphiphilic rod-coil BCPs based on a short hydrophilic flexible segment and a semiconducting low-band-gap copolymer as a rod [28,29,30,31,32,33]. The semiconducting polymer used was (poly[2,6-(4,4-bis-(2-ethylhexyl)-4*H*-cyclopenta[2,1-*b*;3,4-*b’*]dithiophene)-*alt*-4,7(2,1,3-benzothiadiazole)]) (PCPDTBT), responsible for photon absorption and hole transport. Meanwhile, the coil block, constituted by a short chain of poly-4-vinylpyridine (P4VP), can interact with water, thus assuring the colloidal stability of the WPNP dispersions. This helps to control the organization at nanoscale for obtaining peculiar morphologies which influence the device performances. As a matter of fact, in 2018, we reported on PCPDTBT-*b*-P4VP bearing a coil block with five repeating units of 4-vinylpyridine (4VP) which helped to produce a working device with an efficiency up to 2.5% [31].

In this work, we synthetized a new amphiphilic rod-coil BCP bearing PTB7 as the rod block and a segment of P4VP constituted by around 15 repeating units. PTB7 was selected as the rod block because of the higher polymer crystallinity with respect to PCPDTBT, which led to higher efficiency in OPV devices [34]. The rod-coil PTB7-*b*-P4VP was deeply characterized with UV-visible (UV-vis) and Fourier transform infrared (FTIR) spectroscopies, proton nuclear magnetic resonance (^1^H-NMR) spectroscopy, bidimensional NMR (2D-NMR), and size exclusion chromatography (SEC). Matrix-assisted laser desorption/ionization time-of-flight mass spectrometry (MALDI-TOF MS) and differential scanning calorimetry (DSC) were also performed. The surface energy of PTB7-*b*-P4VP was also measured. We investigated its ability to form WPNPs in blend with PC_71_BM. The aqueous suspensions were analyzed through dynamic light scattering (DLS) and transmission electron microscopy (TEM). Finally, they were employed to realize WPNP-based films, studied with atomic force microscopy (AFM) and grazing incidence X-ray diffraction (GIXRD), and subsequently tested as an active layer in sustainable NP-OPV devices.

In this study, we succeeded in obtaining WPNPs with a morphology suitable to split the exciton and generate free charges. The TEM images revealed that the WPNPs were spherical with a Janus inner morphology (also known as biphasic) [35,36]. The Janus morphology suits photovoltaic applications much better than the core–shell one, which minimizes the surface area between the donor and acceptor materials by decreasing the efficiency of the NP-OPVs [19,37].

## 2. Materials and Methods

### 2.1. Materials

All reagents were purchased from Sigma-Aldrich Italia and Ossila BV (Leiden, The Netherland) and used as received. Commercially available 4-vyilpyridine (4VP) and anisole, used to prepare the coil block, were distilled on calcium hydride under a reduced pressure and stored at −20 °C under nitrogen atmosphere. The coil block P4VP with 15 repeating units was synthetized as reported [36]. 2,2,5-trimethyl-4-phenyl-3-azahexane-3-nitroxide (TIPNO) was purchased from Sigma-Aldrich Italia and handled under nitrogen atmosphere. Brominated tert-butyl isopropyl phenyl nitroxide (TIPNO-PhBr) was synthesized as reported [38]. All other solvents used for the chemical reactions were dried by standard procedures. All manipulations involving air-sensitive reagents were performed under dry nitrogen atmosphere. 1,2-dideutero-1,1,2,2-tetrachloroethane (TCE-d2) was purchased from TCI Chemicals Europe N.V. for the NMR spectrum.

For WPNP production through the miniemulsion approach, MilliQ water grade ultrapure was used (resistivity of ∼18 at 25 °C). Clevios AI 4083 PEDOT:PSS was purchased from Heraeus (Hanau, Germany). PC_71_BM, PC_61_BM, and PC_60_BM-PEG were obtained from Solenne BV (Groningen, The Netherlands) and used as received.

### 2.2. Synthetic Procedures

#### 2.2.1. Synthesis of PTB7-*b*-P4VP

Synthesis of the macromer PTB7. 4,8-bis[(2-ethylhexyl)oxy]-2,6-bis(trimethylstannyl)benzo[1,2-*b*:4,5-*b’*]dithiophene (BDTOEHSn) (80 mg, 0.103 mmol) and 2-ethylhexyl 4,6-dibromo-3-fluorothieno[3,4-*b*]thiophene-2-carboxylate (FTThBr) (50 mg, 0.103 mmol) were dissolved in a mixture of 1 ml of toluene and 300 μL of dimethylformamide (DMF) in a dry, oxygen-free Schlenk tube. After that, Pd_2_(dba)_3_ (1.9 mg, 2.06 × 10^−3^ mmol) and P(*o*-tol)_3_ (2.5 mg, 8.24 × 10^−3^ mmol) were dissolved in 170 μL of toluene and added to the reaction mixture. Several freeze-pump-thaw cycles were performed to remove any remaining oxygen trace. The mixture was heated at 120 °C under stirring. The color turned quickly from dark red to dark blue. After 24 h, the reaction mixture was cooled at room temperature and a sample was taken, filtered through Celite® and characterized by ^1^H-NMR, SEC, MALDI-TOF MS, UV-vis, and FTIR techniques. The ^1^H-NMR spectrum (Figure 1) is consistent with that reported in the literature [39,40,41].

Coupling between the rod PTB7 and the coil P4VP. After taking a sample of the macromer PTB7 for the characterization, the coil block of about 15 repeating units of P4VP (30 mg), synthetized and characterized as reported [33,42], was dissolved in 1 mL of toluene and 300 μL of DMF. The obtained mixture was added to the reaction mixture at room temperature. Next, 200 μL of catalytic solution was added to ensure the coupling between the two blocks. The reaction mixture was heated again at 120 °C for 24 h. Then, the Schlenk tube was cooled at room temperature and 200 μL of degassed 2-bromothiophene was introduced to cap the residual stannyl terminal groups. After 24 h at 120 °C, 200 μL of degassed 2-(tributylstannyl)thiophene was added to cap the residual bromide terminal groups. Then, 200 μL of fresh catalytic solution was added during each capping. After 24 h, ~70 mg of scavenger resin (3-mercaptopropyl-functionalized silica gel) was added, and after about 2 h under stirring at 120 °C, the mixture was filtered through Celite®, washing with toluene and chloroform. After being concentrated in a vacuum, the filtrate was poured into methanol to remove the methanol-soluble homopolymer P4VP. Subsequently, the product was dissolved in a small quantity of chloroform and poured again into hexane (solvent selective for the rod block) to obtain a blue solid. Finally, the so-obtained material was filtered and extracted sequentially with Soxhlet apparatus in acetone, hexane, and chloroform. The final product appeared as a dark blue solid.

#### 2.2.2. Synthesis of PC_71_BM: PTB7-*b*-P4VP (1.1:1) Blend WPNPs

The PC_71_BM:PTB7-*b*-P4VP (1.1:1) blend WPNPs were prepared through a modified miniemulsion approach, based on the procedure developed by Landfester [15,43]. In a typical experiment, the two active materials composing the blend were dissolved into a mixture of toluene and *o*-xylene (in ratio 50:50) in order to achieve two starting solutions with standard concentrations, i.e., 20 mg∙mL^−1^ and 10 mg∙mL^−1^ for PC_71_BM and PTB7-*b*-P4VP, respectively. A proper quantity of each starting organic solution was mixed up to obtain an active material organic solution with actual ratio of PC_71_BM:PTB7-*b*-P4VP = 1.1:1. Then, 100 µL of the blend organic solution was sonicated and slowly poured into 1 mL of MilliQ water pre-heated at 50 °C, under vigorous stirring. After ~45 min, the macroemulsion was sonicated for 40 min at 50 °C in an ultrasonic bath to achieve a stable miniemulsion that was heated up to 105–110 °C in order to remove the solvent mixture under a gently stirring. Next, 100 µL of MilliQ water pre-heated at 50 °C was added to restore the aqueous phase evaporated with the organic solvents. After ~2 h, a dark blue-brown aqueous suspension of WPNPs was obtained. All steps of the preparation were performed in air.

### 2.3. Sample Characterization

#### 2.3.1. Nuclear Magnetic Resonance (NMR)

NMR experiments were carried out on a Bruker 500 MHz DM Avance II spectrometer (Bruker Corp., Billerica, MA, USA) operating at 11.7 T, equipped with a 5 mm probe whit gradient unit on z axis, and thermostated at 330 K. ^1^H acquisition parameters: 90°, pulse 9.10 µs, spectral width 5240 Hz, and number of transient 1024. ^1^H-NMR spectra were recorded at 330 K using 1,2-dideutero-1,1,2,2-tetrachloroethane (TCE-d2 as the solvent; all chemical shifts were reported in the standard notation of parts per million (ppm) using the peak of residual proton signal of TCE-d2 (^1^H = 5.94 ppm) as the internal reference.

Diffusion NMR experiments were performed with a pulsed-gradient stimulated echo sequence, using a bipolar gradient. Sequence delays of Δ = 100 ms (diffusion delay) and a LED delay of 50 ms were used.

For each experiment, rectangular PFGs, with a duration of 1 ms and a recovery delay of 100 us, were applied. The z axis gradient strength was logarithmically incremented in 32 steps from 2% up to 98% of its maximum value. After Fourier transformation and baseline correction, the experiments were processed using the Bruker TopSpin software package (4.0.6 version) (Bruker Corp., Billerica, MA, USA). 

A two-dimensional total correlation spectroscopy (TOCSY) experiment was acquired by applying a phase-sensitive Bruker library sequence, with Echo/Antiecho-TPPI gradient selection and a TOCSY spinlock mixing time of 90 ms.

#### 2.3.2. Size Exclusion Chromatography (SEC)

SEC measurements were carried out by using an integrated Waters Alliance GPCV2000 SEC system (Waters Corp., Milford, MA, USA) equipped with two on-line detectors: (1) a differential viscometer (DV); and (2) a differential refractometer (DRI) as a concentration detector. As the chemical composition of the PTB7-*b*-P4VP is very complex, different SEC experimental conditions were used, particularly changing the eluent and the column set. The experimental conditions are summarized in the Appendix A.

#### 2.3.3. Matrix-Assisted Laser Desorption/Ionization Time-of-Flight Mass Spectrometry (MALDI-TOF MS)

MALDI-TOF MS measurements were carried out in reflectron mode to record spectra by means of a 4800 Proteomic Analyzer MALDI-TOF/TOF instrument (Applied Biosystems, Foster City, CA, USA) equipped with a Nd:YAG laser at a wavelength of 355 nm, a <500 ps pulse, and a 200 Hz firing rate. The accelerating voltage was 15 kV. External calibration was performed using an Applied Biosystems calibration mixture consisting of polypeptides with different molecular weight values. The irradiance was maintained slightly above the threshold to obtain a mass resolution of about 6000–8000 fwhm. Mass accuracy was about 50 ppm. The best mass spectra were recorded using trans-2-[3-(4-terbutylphenyl)-2-methyl-2-propenylidene]malononitrile (DCTB) as the matrix.

#### 2.3.4. Differential Scanning Calorimetry (DSC)

DSC measurements were performed by means of a TA Instrument Q100 DSC (Mettler Toledo, Columbus, OH, USA) calibrated with melt purity indium standard (156.6 °C and 28.45 J/g). Before any experiment, the baseline was recorded using empty aluminum pan (reference and sample). About 3–4 mg samples were used. Each sample was analyzed under nitrogen atmosphere (a flow rate of 50 mL/min) using the following runs: (a) heating at 10 °C/min from −50 to 200 °C; (b) cooling at 50 °C/min from 160 to −90 °C; and (c) heating at 10 °C/min from −50 to 200 °C. Three repeated cycles were performed to verify the reproducibility of thermal transitions. The glass-transition (*T*_g_) temperatures measured in the second heating run were computed by the midpoint method.

#### 2.3.5. Fourier Transform Infrared Spectroscopy (FTIR)

FTIR experiments were performed using a Spectrum One FTIR spectrometer (Perkin–Elmer, Waltham, MA, USA) equipped with a deuterated triglycine sulphate (DTGS) detector. The spectral resolution used for all experiments was 4 cm^−1^. For attenuated total reflectance (ATR) measurements, the internal reflection element (IRE) was a three-bounce 4 mm-diameter diamond microprism. Cast films were prepared directly onto the internal reflection element by depositing the solution of interest (~5 μL) on a KBr disk, allowing the solvent to completely evaporate.

#### 2.3.6. UV-Visible Spectroscopy

UV-visible absorption spectra were recorded with a Lambda 900 spectrometer (Perkin-Elmer, Waltham, MA, USA). Optical characterizations were performed on the polymers dissolved in chloroform, the aqueous suspensions, and the device active layer.

#### 2.3.7. Contact Angle and Surface Energy Measurements

PTB7 and PTB7-*b*-P4VP films were deposited by blade coating (v = 10 mm s^−1^, 50 °C) on glass substrates from 8 mg∙mL^−1^ and 2 mg∙mL^−1^ chlorobenzene solutions, respectively. After this, the deposition the films were subjected to annealing treatment at 120 °C for 5 min. Contact angles (CA) were measured using a Drop shape analyzer DSA30 (KRUSS, Hamburg, Germany) in the sessile static mode [44]. About 10 independent drops of solvent were dropped on each substrate, and left and right contact angle values were extrapolated by a circle fitting algorithm. For each drop, 15 CA estimations were performed, and the average value was calculated.

#### 2.3.8. Dynamic Light Scattering (DLS)

The hydrodynamic diameter, polydispersity, and ζ potential of WPNPs were determined through DLS using a Brookhaven 90 Plus size analyzer (Holtsville, NY, USA). The apparatus was equipped with a He-Ne laser which emitted light at λ = 632.8 nm and a detector which recorded intensity at a fixed scattering angle of θ = 90°. All measurements were performed at room temperature. Samples for the measurements were prepared by properly diluting the original WPNP suspensions with MilliQ water.

#### 2.3.9. Transmission Electron Microscopy (TEM)

TEM and energy-filtered TEM (EFTEM) images were collected with a 200 kV ZEISS LIBRA 200 FE (Carl Zeiss Microscopy, Oberkochen, Germany) equipped with a second-generation column Ω filter, and the EFTEM images were recorded by centering the energy-selecting slit at 17 eV and 30 eV with a ±2 eV range. The four samples were prepared using the following procedure. The mother solution was diluted 1:3 with MilliQ water, 7 µL of WPNP suspension was dropped on a copper grid, and the excess water was blotted with filter paper after 1 min. The WPNP diameter was measured using Olympus, a TEM imaging platform, and dm = ∑dini/∑ni, where ∑ni is the number of particles [45].

#### 2.3.10. Atomic Force Microscopy (AFM)

AFM was performed with a commercial equipment (AFM, NT-MDT NTEGRA Spectrum Instruments, Moscow, Russia) in tapping mode with a cantilever NSG10 operating at a typical resonance frequency of 140–390 kHz. The samples were prepared using glass slides and were properly treated with plasma etching for 10 min before eight repeated depositions of the aqueous WPNP suspensions. The measurements were performed before and after annealing of the samples at 90 °C for 20 min in air. An active layer of devices was also analyzed through AFM.

#### 2.3.11. Grazing Incidence X-ray Diffraction (GIXRD)

GIXRD measurements were performed at the X-ray diffraction beamline 5.2 using Elettra, a synchrotron radiation facility in Trieste (Italy). The X-ray beam emitted by the wiggler source on the Elettra 2 GeV electron storage ring was monochromatized by a Si(111) double-crystal monochromator, focused on the sample and collimated by a double set of slits giving a spot size of 0.2 × 0.2 mm. The beam was monochromatized at 1.4 Å. The samples were oriented by means of a four-circle diffractometer with a motorized goniometric head (internally developed). The diffractometer meant that sample surface alignment could be carried out in the horizontal plane containing the X-ray beam by means of laser light reflection. For every sample, seven images at variable incidence (from −0.1° to 0.2°, step 0.05°) were taken, each one by rotating the sample of 360° around the normal to the surface in 60s of exposition to the beam. The bidimensional diffraction patterns were recorded with a 2M Pilatus silicon pixel X-ray detector (DECTRIS Ltd., Baden, Switzerland) positioned perpendicular to the incident beam, at a distance of 350 mm from the sample. Patterns were calibrated by means of a LaB6 standard and integrated using the software fit2d, obtaining powder-like patterns, corrected for geometry, Lorentz, and beam polarization effects, in the azimuthal region between 70° and 110° for the OOP signal and between 170° and 180° for the IP signal. Peak positions were extracted using the Win Plot program. Bidimensional images, representing the intensity as a function of qxy and qz, where q is the reciprocal lattice vector (or transferred momentum) expressed in Å^−1^, were obtained with GIDVis software, version 3 (Mathworks, Natick, MA, USA).

### 2.4. OPV Device Fabrication and Characterization

Glass substrates (25 × 25 × 1.1 mm^3^) coated with indium tin oxide (ITO) with a sheet resistance of 15 Ω sq^−1^ were used to fabricate the devices. After mechanical cleaning with fibreless paper, the substrates were sonicated at 50 °C in water, acetone, and 2-propanol for 10 min each step. After being dried with a N_2_ gun, the substrates were subjected to plasma treatment for 10 min before the deposition of a layer (~20 nm) of poly(3,4-ethylenedioxythiophene):polystyrene sulfonate (PEDOT:PSS) (~10 drops) at 1500 rpm for 60 s. The PEDOT:PSS layer was annealed at 200 °C for 15 min through a hotplate under nitrogen flux. After this, the PEDOT:PSS layer was exposed to UV–O_3_ treatment for 10 min to make it more hydrophilic. Then, 200 μL of ethanol was added to the WPNP suspension to facilitate the deposition. Next, the WPNPs were spin-coated on the PEDOT:PSS layer at 500 rpm for 40 s. Eight repeated depositions of WPNP suspension were needed to provide an active layer with an absorbance suitable for the device operation. After each deposition, the substrate was heated on a hotplate at 60 °C for few min to completely remove water, and then washed with ethanol to make the following deposition as easy as possible. A thin layer (~20 nm) of phenyl-C61-butyric acid methyl ester (PC_61_BM) was deposited at the top of the active layer from a dichloromethane solution by spin-coating at 4000 rpm for 10 s. A final annealing treatment was carried out at 90 °C for 20 min in air. The finalized active layers were then annealed at 90 °C for 20 min in air.

The OPV devices obtained with standard deposition of the active layers from organic solvents were prepared inside the glovebox. PTB7-*b*-P4VP and PC_71_BM were dissolved in chlorobenzene at a composition 1:1.5 weight ratio, with a solute concentration of 25 mg∙mL^−1^. This solution was stirred overnight at 65 °C. Next, 10 min after adding 2% *v*/*v* anisaldehyde, the blend was spin-coated at 1500 rpm for 60 s. The samples were left at room temperature for 10 min and then annealed at 65 °C for 10 min.

All the substrates were inserted into a glovebox where 10 nm of Ca and 100 nm of Al were evaporated on top of the samples through a shadow mask under a pressure of 1.5 × 10^−6^ mbar. The deposition rates were 0.7 nm∙s^−1^ for Al and 0.1 nm∙s^−1^ for Ca. On each substrate, six devices were separately connected and characterized, each one with an active area of 6.1 mm^2^.

Current density-voltage (J-V) measurements were performed with a Keithley 2602 source meter (Keithley Instruments, Cleveland, OH, USA), under AM 1.5G solar simulation (ABET 2000). The incident power, measured with a calibrated photodiode (Si cell + KG5 filter), was 100 mW∙cm^−2^. The external quantum efficiency (EQE) spectral responses were recorded by dispersing an Xe lamp through a monochromator, using a Si solar cell with a calibrated spectral response to measure the incident light power intensity at each wavelength.

## 3. Results and Discussion

### 3.1. Synthesis and Characterization of the Macromer PTB7

The low-band-gap polymer PTB7 was chosen as the *p*-type semiconducting rigid block. As shown in Figure 1, the macromer PTB7 was synthetized through Stille coupling between the electron-rich monomer BDTOEHSn (1) and the electron-poor FTThBr (2) [40], using Pd_2_(dba)_3_/P(*o*-tol)_3_ as the catalytic system. The macromer PTB7 (3) was fully characterized from molecular, thermal, and spectroscopic points of view. The ^1^H-NMR (Figure 1) and the FTIR spectra (Appendix A) were in agreement with the data reported in the literature [39,40,41]. The UV-vis spectrum of the polymer dissolved into chloroform is depicted in Figure 2. It shows a broad absorption peak with a maximum at ~600 nm, which is typical of the PTB7 backbone [40,46]. Macromer molecular weight values, Mn and Mw, were determined by size exclusion chromatography (SEC) at 14,380 g mol^−1^ and 56,350 g mol^−1^, respectively. The molecular weight was confirmed by means of MALDI-TOF MS spectrometry (Appendix A). The end-groups of polymer chains were also determined. Among them, hydroxyl groups from terminal stannyl group hydrolysis and even chains bearing a tin atom were observed. These analyses were performed on the crude samples without any solvent extractions, as the macromer PTB7 could not be extracted in order to preserve the reactivity of the end-groups for the following coupling with the brominated coil block. The relatively low molecular weight obtained for the macromer represents a compromise between its length and its polydispersity. Narrow polydispersity is required to have homogeneous aqueous suspensions suitable for the OPV active-layer deposition [47].

### 3.2. Synthesis and Characterization of the Rod–Coil Block Copolymer PTB7-b-P4VP

PTB7-*b*-P4VP was synthetized using step-growth-like polymerization method. We prepared and characterized the two properly functionalized polymeric backbones, PTB7 and P4VP, for the following coupling [33,48]. The hydrophilic segment P4VP was tailored with around 15 repeating units on the basis of the relationship between the molecular structure and the NP-OPV devices efficiency found for the PCPDTBT-based rod-coil BCPs with P4VP segments of increasing length [28]. Furthermore, we kept in mind that longer alkyl chain surfactants improve the charge mobility in OFET based on NPs [49]. In addition to this structural consideration, the design of PTB7-*b*-P4VP took into account the process feasibility, in order to obtain a shorter coil, and a chain-growth-like approach was needed with a considerable increase in the synthetic complexity of the material [33]. The coil presence avoids the use of surfactants as it interacts with the aqueous medium and stabilizes the aqueous-non-aqueous interfaces which ensure colloidal stability of WPNP dispersions [30]. The segment of P4VP (4) was achieved through nitroxide-mediated radical polymerization (NMRP) using the radical TIPNO as a mediator, as reported in the literature [50]. The coil length was confirmed by ^1^H-NMR analysis and the calculated repeating unit number was considered as an average value. The two blocks were linked together through Stille coupling, as shown in Figure 1. The purification to remove the unreacted P4VP was performed through repeated precipitations into methanol (coil-selective solvent) [27]. The resulting PTB7-*b*-P4VP (5) was structurally and optically characterized.

The UV-vis absorption spectrum of the PTB7-*b*-P4VP was compared with the spectrum of a homemade PTB7 polymer. In the UV-vis spectra of the samples dissolved into chloroform (Figure 2), it is possible to notice that the PTB7-*b*-P4VP spectrum retains the PTB7 shape, showing a broad absorption band with a maximum at ~600 nm, which is typical of PTB7 [40,46]. The blue shift in the absorption spectrum occurs due to the different molecular weights of the samples. A mild scattering is detectable for the block copolymer PTB7-*b*-P4VP due to the use of chloroform as the solvent for the spectrum acquisition, which is a good solvent for the rod block and an adequate compromise for the coil one [27].

Otherwise, FTIR analysis (Appendix A) shows some differences with respect to PTB7 one and additional signals related to the coil block [50]. In particular, in the FTIR spectrum of PTB7-*b*-P4VP, the peaks between ~1400 and 1500 cm^−1^ are broadened compared to those of PTB7 as the P4VP segment bands in this region at 1558 cm^−1^ and 1417 cm^−1^, assigned to C=C and C=N vibrations at 1597 cm^−1^ and 1493 cm^−1^, respectively, ascribable to the C=C modes of the aromatic rings, overlap with PTB7 signals. Furthermore, at around 800 cm^−1^, a signal is observable due to C–H vibrations of P4VP.

Several differences between the macromer PTB7 and the rod–coil PTB7-*b*-P4VP samples were observed in the ^1^H-NMR experiments too. In Figure 1a, ^1^H-NMR spectra of the macromer PTB7 and of the PTB7-*b*-P4VP are displayed. As already cited, the spectrum associated to PTB7 is in agreement with the literature, while PTB7-*b*-P4VP shows some dissimilarities. Particularly, as indicated by red arrows in Figure 1a, in the insertion corresponding to the zoom on the aromatic region, we observe several differences ascribable to the presence of the TIPNO and the coil segment. A new broadened peak appears at 6.58 ppm, while the signals in the 7.0–7.2 ppm and 7.55–7.70 ppm ranges helped to modify the shape and intensity as a consequence of the P4VP attachment. This evidence denoted the actual presence of the coil segment into the analyzed sample, but other experiments were required to confirm the effective coupling between the two blocks. 

At this purpose, we entrusted 2D-NMR techniques, and especially diffusion ordered spectroscopy (DOSY), which helps to resolve a mixture of compounds based on their diffusion coefficients which are dependent on the size and shape of the molecules, as well as total correlation spectroscopy (^1^H-^1^H TOCSY—Appendix A), which highlights correlations between all protons within a given spin system. The DOSY spectrum of PTB7-*b*-P4VP is depicted in Figure 1b. It exhibits several signals related to distinct aliphatic segments which diffuse together, denoting that the two blocks are attached.

The molecular weight distributions (MWDs) of the macromer PTB7 and the rod–coil PTB7-*b*-P4VP were determined by means of size exclusion chromatography (SEC). In order to compare the two samples, the MWDs were determined using a mixture of solvents, THF:DMF = 80:20 [33], as the eluent, as reported in Table 1 (Appendix A). Unfortunately, P4VP was insoluble in this mixture, and its MDW was determined using DMF as the mobile phase (Appendix A). The MWD of P4VP was very narrow, revealing the very low polymerization degree of the sample (10–15 repeating units), in agreement with NMR analysis. The observed variation of the molecular weight between the macromer PTB7 and the rod–coil PTB7-*b*-P4VP was very small (around 10%), but was consistent with the addition of the short coil block.

The clear-cut proof of the covalent bond formation between the macromer PTB7 and the coil P4VP was provided by differential scanning calorimetry (DSC). Indeed, the thermogram depicts two distinct transitions: one at ~120 °C, attributable to the coil block [30,51], and another one at ~170 °C, related to the rod segment [52,53,54] (Appendix A).

The linkage of the coil segment to PTB7 also influences polymeric interactions with substrates, as revealed by the contact angle and surface energy measurements. Contact angles were determined using diiodomethane (DIM) (γD = 50.8 mN m^−1^; γP = 0) [55] and acetonitrile (ACN) (γD = 20.8 mN m^−1^; γP = 8.5 mN m^−1^) as test liquids (Appendix A) [56]. The measured values were used to calculate the dispersive and polar components of the surface energy, according to the Fowkes model [57]. The results are reported in Table 2.

The surface energy of PTB7 is 33.1 mN m^−1^, in agreement with the typical values (20–35 mN m^−1^) reported in the literature for other conjugated polymers using OPV applications, with a very low polar contribution [58,59,60].

The addition of the short hydrophilic P4VP block (corresponding to about 10% of variation of *M*_n_, according to SEC data) to the conjugated polymer structure leads to a slight increase in the polar component of the surface energy, from 0.06 mN m^−1^ to 0.5 mN m^−1^.

### 3.3. Synthesis, Characterization, and Deposition of the Water-Processable Nanoparticles

WPNPs were prepared with the blend made of [6,6]-phenyl-C71-butyric acid methyl ester (PC_71_BM) and amphiphilic PTB7-*b*-P4VP in ratio 1.1:1 by using a miniemulsion procedure (Figure 3). In order to avoid the use of chlorinated solvents in the OPV device fabrication, a mixture of toluene and *o*-xylene (in ratio 50:50) was used to dissolve both active materials during the process. The complete removal of halogenated solvents in the WPNP production, even in the dissolution step, improves the sustainability of the whole process. In fact, the employment of a toluene: *o*-xylene mixture, i.e., organic solvents, supports the scale-up of the WPNP production process. It is important to remark that a controlled separation of organic solvents from the aqueous suspension can be integrated into a proper circular industrial plant equipped with condenser systems to recover and reuse the organic solvents.

The mixture of chosen organic solvents was adapted to obtain high-concentration WPNP suspensions, since the use of pure toluene led to the loss of a high amount of the blend during the suspension preparation as a result of the material aggregation. The concentration of the obtained suspensions was extrapolated from the absorbance value of the correlated WPNP aqueous suspension (Appendix A). The UV-vis absorption spectrum shows the typical profiles of the two materials constituted by the blend and presents a broad peak at ~600 nm associated to PTB7. All the peaks are broadened and slightly shifted with respect to PTB7-*b*-P4VP and PC_71_BM organic solutions. This evidence is attributable to the nanoaggregation within the WPNPs which occurs during the miniemulsion process.

The WPNPs were analyzed through DLS to determine their size and stability in aqueous medium. The size distribution data, expressed by number, reveal a mean hydrodynamic diameter (d_H_) of 79.2 ± 1.3 nm. The determination of the d_H_ in these kind of samples is almost complex. As a matter of fact, the multimodal size distribution (MSD) indicated that the aqueous suspension is constituted by two main populations: the former centered at about 50 nm and the latter centered at about 200 nm (Appendix A). The ζ-potential value of −46.91 ± 0.68 mV indicates good colloidal stability of the aqueous suspension.

With the aim to deeply investigate the WPNP morphology and identify their internal organization, TEM experiments were performed. Although the images are affected by blurred contours (maybe caused by *o*-xylene residual in the samples), they reveal a spherical shape of WPNPs and demonstrate the presence of NP aggregates. There are larger NPs surrounded by smaller nanostructures, in agreement with DLS results. 

By exploiting the conventional TEM (CTEM) images, it was possible to estimate the mean size of the WPNPs and of the internal core too. The energy-filtered TEM (EFTEM) was used to acutely investigate the WPNP internal structure, composition, and domain distribution. The images collected with EFTEM are generated by the electrons with a specific and selected energy loss. Selecting the correct energy loss is mandatory in order to acquire the electron energy loss spectroscopy (EELS) spectrum of each studied sample. In the case of the samples analyzed in this work, the EELS spectrum area of interest is the low-loss zone which corresponds to the plasmon signal (Figure 4f). PTB7-*b*-P4VP and PC_71_MB have different electronic density; therefore, as established from the current literature, the maximum PC_71_BM plasmon occurs at 25 eV [28,30,31,61], while the maximum PC_71_BM:PTB7-*b*-P4VP blend occurs at around 22–23 eV. The EFTEM images were collected at 17 eV and 30 eV in order to improve the contrast and to completely cut off the PC_71_BM contribution to the image in order to highlight the polymer localization in the first case and vice versa in the second one. Therefore, the PC_71_MB was darker in the image taken at 17 eV (Figure 4b), and was lighter in the image taken at 30 eV. However, in the case of PTB7-*b*-P4VP, results showing opposite levels of brightness were observed (Figure 4c). Comparing the EFTEM images collected at different energy losses, it was evident that PTB7-*b*-P4VP segregates at one side of the WPNPs (Figure 4b) and a PC_71_BM-rich region segregates to the other side of the WPNPs (Figure 4c) [62].

This evidence refers to a Janus-like NP morphology. We also performed an accurate EDX analysis to effectively describe the PTB7-*b*-P4VP assembly inside the nanostructure. The line scan map, reported in Figure 4e, shows the element concentration profiles along the yellow line which cross a NP in the STEM image (Figure 4d). By comparing the sulfur and carbon concentration profiles, it was evident that the sulfur did not have homogeneous distribution inside the WPNP. This suggests that the sulfur-enriched domain was formed during WPNP synthesis. This fact supports the idea that the self-assembly of amphiphilic block copolymers influenced the nanosegregation of the donor acceptor domains, as highlighted by the EFTEM images [63].

Since the stable aqueous suspensions of PC_71_BM:PTB7-*b*-P4VP were subsequently applied for the fabrication of the OPV active layer, preparatory deposition optimization was required. After several attempts, we established that multiple depositions were necessary to prepare an active layer with absorbance that was high enough to obtain working devices (at least 0.2). In particular, on a properly treated ITO substrate, a layer of PEDOT:PSS was deposited (as hole transporting layer) and annealed at 200 °C for 15 min. This layer was exposed to ultraviolet ozone (UV-O_3_) treatment to obtain a uniform coverage, thus overcoming its wettability issues with aqueous suspensions [31,64,65].

The AFM study of the as-deposited film was carried out (Figure 5). The AFM images before annealing treatment (Figure 5a,b) revealed that the coverage of the substrate was compact, but the film was inhomogeneous. A thermal annealing at 90 °C for 20 min was required to improve the film quality (Figure 5c,d). Indeed, the root mean square (RMS) roughness value decreased from 16.1 nm (before annealing) to 11.3 nm (after annealing), while the thickness (including the PEDOT:PSS layer) decreased from 90 nm to 80 nm. In fact, the phase images (Figure 5b,d) follow the edges of all the features present in the corresponding height images, thus confirming the sensibly higher compactness and homogeneity after annealing treatment.

The film was also studied through grazing incidence X-ray diffraction (GIXRD) to investigate its crystallinity degree. The comparison between 2D images taken at an incident angle close to 0.0° and at 0.15° denotes a larger order close to the substrate, as often observed for polymeric materials (Appendix A) [66,67]. On the other hand, no periodic repetition was observed for the presence of the fullerene materials. We can assume that PTB7-*b*-P4VP preserves its semicrystalline nature, even if deposited from WPNPs. Otherwise, PC_71_BM does not aggregate in a coherent manner as PC_61_BM instead does. This is due to the PC_71_BM shape, which is less spherical than PC_61_BM and, thus, is harder to induce to aggregate, even with thermal treatment.

### 3.4. Device Fabrication and Characterization

#### 3.4.1. Device Architecture Optimization: Choosing the Electron Transporting Layer (ETL)

To investigate the sustainable fabrication of NP-OPVs using PTB7-*b*-P4VP, WPNP-based devices were prepared with a direct configuration. The active layer was prepared by eight sequential spin-coated depositions of PC_71_BM:PTB7-*b*-P4VP (1.1:1) blend WPNPs on top of an ITO/PEDOT:PSS anode electrode, using the process conditions previously described. Finally, a Ca-Al cathode electrode was evaporated on top. As demonstrated by the current density-voltage (J-V) curves reported in Appendix A, the devices are under short circuit. This implies that, even after thermal treatments, the compactness or homogeneity of the WPNP active layer is still not sufficient to avoid shunts. 

To reduce the electric leakage, the device architecture was modified by depositing a thin layer of fullerene molecules on top of the WPNP blend film [31,68]. To completely avoid processing from chlorinated solvents, the deposition of a fullerene derivative soluble in alcoholic medium, i.e., the polyethylene glycol-modified fullerene (PC_61_BM-PEG), was firstly considered. This fullerene was reported to work effectively as a solution processable cathode interlayer for polymer solar cells [69].

As displayed in the Appendix A, the devices prepared with PC_61_BM-PEG were under short circuit. PC_71_BM and PC_61_BM were then tested; following the literature, a small amount of dichloromethane was used for processing [31,68]. In the case of PC_71_BM, there was a slight improvement, but a large number of shunts still affected the device (Appendix A). A significant improvement was obtained when using PC_61_BM on top of the WPNP blend film. In this case, the J-V curves (Appendix A) demonstrated a relevant reduction in electrical or short-circuit leakages, suggesting that PC_61_BM can fill the voids in the WPNP layer to prevent shunts. In addition, relatively uniform photovoltaic characteristics were obtained (Appendix A). This was in accordance with the AFM images (Figure 6), showing a smoother and more homogeneous surface upon PC_61_BM deposition, with a roughness that is considerably reduced to 5.6 nm.

#### 3.4.2. Comparison between Device from Aqueous Suspension and Conventional Bulk Heterojunction (BHJ) Device

The photovoltaic characteristics of the optimized ITO/PEDOT:PSS/ WPNP/PC_61_BM/Ca/Al device (see Experimental for fabrication details) are reported in Figure 7 and Appendix A, and Table 3. The PTB7-*b*-P4VP was also applied as a donor polymer for comparison in a conventional BHJ solar cell processed through organic solvent. In this case, the active layer was prepared by deposition of a PC_71_BM:PTB7-*b*-P4VP (1.5:1) chlorobenzene solution (device 2).

The device obtained from the aqueous suspension (device 1) performed better than the conventional blend solar cell (device 2), reaching PCE values of 0.85% and 0.63%, respectively. In fact, from the J-V curves and the corresponding PV parameters reported in Figure 7a and Table 3, it can be seen that device 1 has a lower open circuit voltage (V_OC_) with respect to device 2 (0.67 V and 0.63 V), probably because both PC_71_BM and PC_61_BM function as acceptor components [70]. The WPNP-based device exhibits higher short-circuit current density (J_SC_) than the classical blend device (2.46 and 1.83 mA/cm^2^). As displayed in Figure 6b, the J_SC_ values of the two devices, calculated by combining the EQE spectra with regards to the AM1.5 G solar radiation spectrum, were 2.78 mA/cm^2^ for device 1 and 1.99 mA/cm^2^ for device 2, respectively. These values are reasonably in accordance with those obtained from J-V measurements (Table 3).

The device made from aqueous suspension exhibited a relatively higher fill factor (FF) than the classical blend devices (0.51 and 0.47 in Table 3). It should be noticed that these small FF values suggest an overall non-optimal exciton dissociation or charge collection to the electrodes for the two devices under comparison. To gain some insight on the exciton dissociation and the charge transport and collection, the photocurrent densities (J_ph_) versus the effective voltage (V_eff_) for the two devices are compared in Figure 6c, where J_ph_ is calculated as J_ph_ = J_light_ − J_dark_ and V_eff_ = V_O_ − V where V_O_ is the voltage if J_light_ = J_dark_ and V is the applied voltage [71,72,73]. From Figure 7c, it can be seen that the two devices exhibited rather different J_ph_ − V_eff_ curves. The J_ph_ of the device prepared from aqueous solution, showed a rapid linear increase as a function of V_eff_ at low V_eff_. However, at higher V_eff_, the bias dependence was notably reduced. This indicates that the losses that limit the photogenerated exciton extraction are reduced under a high internal field [74]. This is not the case for device 2, where the J_ph_ still exhibits a relevant bias dependence, even under high internal fields. This is the signature of quite a pronounced hindrance to charge dissociation, transport, and collection which occur in conventional PTB7-*b*-P4VP-based devices.

Figure 6 shows a comparison of the AFM images of device 1 and device 2, as well as a reference. Interestingly, the image of the device 2 (Figure 6c) reveals a non-optimal AFM topography with large macrodomains of 300–600 nm, which is consistent with the pronounced hindrance to charge dissociation and/or transport and/or collection occurring in conventional BCP-based blend device. In the WPNP-based active layer, the nanoparticle dimensions and their Janus morphology observed through the TEM images (Figure 4) likely favors the charge separation. In addition, the presence of PC_61_BM helps to collect the charges at the electrodes. 

The self-assembly of the amphiphilic rod-coil PTB7-*b*-P4VP is useful in order to gain pre-organized nanostructures which are suitable for proper charge dissociation, transport, and extraction [29]. On the other hand, a standard miniemulsion method, providing surfactant use, usually leads to core-shell NPs. It is acknowledged that NPs with a Janus morphology were more suitable for photovoltaic activity compared to the core-shell ones [17,19]. In fact, in a core-shell morphology, the charges generated at the interphase acceptor core-donor shell are entrapped inside the NP and cannot reach the corresponding electrode, while in a Janus one, a moderate continuous conductive path addicted to the NP orientation can be more easily generated. Unfortunately, the pre-determination of the nanodomain morphology is very challenging because it depends on a lot of parameters and conditions, such as the rod and coil block ratio, the donor and acceptor concentrations, their ratio, the organic phase during the miniemulsion preparation, and so on.

To gain further insight into the photovoltaic performances of the synthesized PTB7-*b*-P4VP donor polymer, a commercial PTB7 polymer was taken as a reference in a conventional PC_71_BM:PTB7 (1.5:1) blend device prepared from a chlorobenzene solution [75]. The photovoltaic parameters obtained with the commercial PTB7 are listed in Table 3. The BCP under study led to PCE values which are one order of magnitude lower than the PTB7 reference. It cannot be excluded that this significant difference is partially an effect of the different molecular weights of the PTB7-*b*-P4VP compared to the commercial PTB7. Nonetheless, the studies reported in the literature do not justify such a dramatic drop in performances [76,77].

The AFM images show that the active layer corresponding to device 2 (Figure 6c) was less interconnected than the film obtained from the commercial PTB7 (Figure 6d) and was characterized by delimited and unconnected macrodomains. On the other hand, the reference active layer exhibited smaller and more interconnected domains (Figure 6d). The higher compactness of commercial PTB7 compared to PC_71_BM:PTB7-*b*-P4VP is confirmed by RMS roughness value, decreasing from 18.8 nm (Figure 6c) to 11.5 nm (Figure 6d). Such a different morphology of the active layer can be explained by the different crystallinity degree of the two polymers. The GIXRD experiments demonstrated that the homemade polymers PTB7 and PTB7-*b*-P4VP lead to more crystalline films than the commercial one (Appendix A). Moreover, the presence of the P4VP coil block can lead to a decrease in the PCE too. The polar coil block of PTB7-*b*-P4VP can act as an electron trap in the active layer, thus hindering the charge extraction and lowering the efficiency of the NP-OPV devices [78,79].

To increase the efficiency, the developed approach can be improved by reducing the overall ratio between the coil and the rod blocks or by modifying the BCP molecular structure from a synthetic point of view. Unfortunately, this target cannot be reached through a higher molecular weight in the donor polymer because of synthetic issues. It is not possible to increase the molecular weight of the macromer PTB7, because with longer reaction times, the polymer dispersity becomes wider with a detrimental effect on the proper WPNP production. The synthetic difficulty of shortening the P4VP coil block is discussed in Section 2.3. As an alternative way to implement the device performance, we aim to modify the hydrophilic flexible block, using non-ionic polar and shorter capping segments. Finally, it will be possible to change the device architecture by fabricating OPV devices with inverted configuration, thus overcoming the issue due to the orientation of the hydrophilic coil close to the cathode [80].

## 4. Conclusions

In summary, a new amphiphilic rod–coil block copolymer based on the semiconducting polymer PTB7, covalently linked to a tailored segment of P4VP, was synthetized. The obtained PTB7-*b*-P4VP was fully characterized from spectroscopical and structural points of view. The covalent bond formation between the two blocks was demonstrated with several advanced techniques, which synergistically helped to confirm the effective formation of the rod-coil BCP.

The PTB7-*b*-P4VP was able to prepare water-processable nanoparticles in blend with PC_71_BM in aqueous medium without the use of surfactants. The obtained aqueous suspensions were stable. TEM images revealed that WPNPs are spherical, organized in a Janus morphology with PTB7-*b*-P4VP segregated on one side of the NP and a PC_71_BM-rich domain on the other side. The Janus morphology promotes interfaces between donor and acceptor materials and improves the interconnection between donor and acceptor domains. The NP-OPV devices displayed higher PCE than that of a BHJ device achieved by conventional deposition of PTB7-*b*-P4VP from chlorinated solvent, probably due to the peculiar morphology achieved.

In conclusion, we can successfully obtain an unusual WPNP morphology, which differs from the typical core-shell one, to increase the charge extraction and transport. Nevertheless, the device efficiency is still far from that achieved using the commercial PTB7 processed by organic solvents. The approach proposed in this work needs to be further optimized to increase the performance of the device and resemble that of the benchmark PTB7.

## Data Availability

Not applicable.

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
