# Peer review of "Amphiphilic PTB7-Based Rod-Coil Block Copolymer for Water-Processable Nanoparticles as an Active Layer for Sustainable Organic Photovoltaic: A Case Study"

_polymers, 2022, doi:10.3390/polym14081588_

Round 1

Reviewer 1 Report

Diterlizzi reported synthetized a new rod-coil block copolymer (BCP) based on the semiconducting polymer poly({4,8-bis[(2-ethylhexyl)oxy]benzo[1,2-b:4,5-b′]dithiophene-2,6-diyl}{3-fluoro-2-[(2-21 ethylhexyl)carbonyl]thieno[3,4-b]thiophenediyl}) (PTB7) and poly-4-vinylpyridine (P4VP), tailored to produce water-processable nanoparticles (WPNPs) in blend with phenyl-C71-butyric acid methyl ester (PC71BM). The J-V characteristics of the WPNP-based devices were measured obtaining a power conversion efficiency of 0.85%. The efficiency of the WPNP-based devices was higher than that achieved for the devices fabricated with the PTB7-based BCP dissolved in chlorinated organic solvent. Although the PCE of WPNP-based devices is still low, the topic is very interesting for further development from the environment and human health sides. I recommend this manuscript acceptable for potential publication in Polymer after well addressing the following issues:

  1. In the UV-vis spectra of the samples dissolved into chloroform (Figure 1) it is possible to notice that the PTB7-b-P4VP spectrum retains.. , which should shown in Figure 2, rather than Figure 1. Meanwhile, the caption of figure 2 should be revised, absorption spectra of …
  2. The EQE spectra of organic solar cells should be added in the main text for readers’ better understanding.
  3. The more discussion or the related references on the photocurrent densities (Jph) versus the effective applied voltage (Veff) should be added to support your discussion, such as solar RRL 2021, 5, 2100175, Small 2022, 18, 2104215. The charge transport and collection in the active layers are needed to be discussed. 
  4. The Figure 7 shows a comparison of the AFM images of the devices and the references, please note that the blend films should be investigated, rather than devices. The more discussion on the surface roughness of blend films is necessary.
  5. Even if the Janus morphology resulted more suitable for a proper photovoltaic activity with respect to the core-shell one it is not that simple to obtain. How to understand the “proper photovoltaic activity”? The more detailed information should be provided to support your statement.

Author Response

Reviewer #1

Diterlizzi reported synthetized a new rod-coil block copolymer (BCP) based on the semiconducting polymer poly({4,8-bis[(2-ethylhexyl)oxy]benzo[1,2-b:4,5-b′]dithiophene-2,6-diyl}{3-fluoro-2-[(2-21 ethylhexyl)carbonyl]thieno[3,4-b]thiophenediyl}) (PTB7) and poly-4-vinylpyridine (P4VP), tailored to produce water-processable nanoparticles (WPNPs) in blend with phenyl-C71-butyric acid methyl ester (PC71BM). The J-V characteristics of the WPNP-based devices were measured obtaining a power conversion efficiency of 0.85%. The efficiency of the WPNP-based devices was higher than that achieved for the devices fabricated with the PTB7-based BCP dissolved in chlorinated organic solvent. Although the PCE of WPNP-based devices is still low, the topic is very interesting for further development from the environment and human health sides. I recommend this manuscript acceptable for potential publication in Polymer after well addressing the following issues:

  1. In the UV-vis spectra of the samples dissolved into chloroform (Figure 1) it is possible to notice that the PTB7-b-P4VP spectrum retains… , which should shown in Figure 2, rather than Figure 1. Meanwhile, the caption of figure 2 should be revised, absorption spectra of 2.

The authors thank the Reviewer #1, and we have revised accordingly all the manuscript and highlighted the fixed mistakes.

  1. The EQE spectra of organic solar cells should be added in the main text for readers’better understanding.

We have followed the Reviewer suggestion and we added the EQE spectra to the main text, in Figure 6.

  1. The more discussion or the related references on the photocurrent densities (Jph) versus the effective applied voltage (Veff) should be added to support your discussion, such as solar RRL 2021, 5, 2100175, Small 2022, 18, 2104215. The charge transport and collection in the active layers are needed to be discussed.

Thanks to Reviewer comments we have checked our manuscript and realized that the Jph versus the Veff discussion should be increased in quality, and we revised the manuscript accordingly.

In particular:

-we provided a smoother and more logical passage from the PV parameter characteristics and the discussion of the Jph-Veff curves.

- we realized that, to make it short and simple, in the submitted manuscript the reader could get a misleading message about the quality of the WPNP-based device. In the revised version, we were more careful in underlying that also the WPNP-based device exhibits non-optimal characteristics. We described more precisely the Jph vs Veff behavior of the two devices.

-we followed the Reviewer’s suggestion and we added into the references some recent studies where the Jph vs Veff behavior were discussed (e.g. Solar RRL 2021, 5, 2100175; Small 2022, 18, 2104215 and Adv. Energy Sustainability Res 2021, 2, 2100069).

  1. The Figure 7 shows a comparison of the AFM images of the devices and the references, please note that the blend films should be investigated, rather than devices. The more discussion on the surface roughness of blend films is necessary.

The surface morphologies relative to the three devices (device 1, 2 and reference) reported in Figure 7 were obtained using AFM on the corresponding three “naked” blend films deposited on the previous layers used for the device fabrication, without the deposition of PC61BM layer and the cathode evaporation. For this reason, the results reflected the surface morphologies of the blend films. We agree with the Reviewer that talking about "device" in describing the active layer surface morphologies is misleading. For this reason, we changed the term "device" into "film morphologies" and we modified the manuscript accordingly. Roughness values have been added in the Figure 7 caption and in the manuscript and discussed as follows:

“The AFM images showed that the active layer corresponding to device 2 (Figure 7c) was less interconnected than the film obtained from the commercial PTB7 (Figure 7d) and it was characterized by delimited and unconnected macrodomains. On the other hand, the reference active layer exhibited smaller and more interconnected domains (Figure 7d). The higher compactness of commercial PTB7 compared to PC71BM:PTB7-b-P4VP is con-firmed by RMS roughness value, decreasing from 18.8 nm (Figure 7c) to 11.5 nm (Figure 7d).”

  1. Even if the Janus morphology resulted more suitable for a proper photovoltaic activity with respect to the core-shell one it is not that simple to obtain. How to understand the“proper photovoltaic activity”? The more detailed information should be provided to support your statement.

Thanks to Reviewer comments, we revised our manuscript and, for the sake of clarity, we tried to better explain this claim, modifying the manuscript as follows:

“The use of amphiphilic rod-coil PTB7-b-P4VP able to self-assemble is useful to gain pre-organized nanostructures suitable for proper charge dissociation, transport, and ex-traction [29], while a standard miniemulsion method, providing surfactant use, usually leads to core-shell NPs. It is acknowledged that NPs having a Janus morphology resulted more suitable for a photovoltaic activity with respect to the core-shell ones [17,19]. In fact, in a core-shell morphology the charges generated at the interphase acceptor core/donor shell are entrapped inside the NP and cannot reach the corresponding electrode, while in a Janus one a moderate continuous conductive path addicted to the NPs orientation can be more easily generated. Unfortunately, the pre-determination of the nanodomain morphology is very challenging because it depends on a lot of parameters and conditions, as the rod and coil block ratio, the donor and acceptor concentrations, their ratio, the organic phase used during the miniemulsion preparation and so on.”

Reviewer 2 Report

  1. In Figure 2, the authors fail to label the y-axis values. It would be difficult to make comparisons.
  2. In Figure 2, the absorptions only captured to 850nm. However, it seems that both of the figure are not shown zero value in high wavelengths. I suggest that the absorption spectra should be remeasured to a higher wavelength to check if there're differences between two polymers.
  3. In page 10 line 458, the authors prepared the blend of PCBM and PTB7-b-P4VP with a ration of 1.1:1. Is there any reason why the author use this ratio of the blend?
  4. In figure 5, the authors provide AFM in height images and calculate the roughness of both samples. Do the authors find any difference in their phase images? 
  5. Some typos such as page 14 line 588 VOC should be corrected to Voc; Page 14 line 590 JSC should be Jsc. The typos should be checked again in the whole manuscript. 
  6.  The PCEs of the blend in WPNPs and chlorobenzene have little difference (one show 0.85% and the other show 0.63%). The one in WPNPs appears to have higher Jsc than the one in chlorobenzene. I suggest the authors to check again with external quantum efficiency (EQE) whether there is a notable difference in both devices.

Author Response

(The authors gave the same response as above.)

Round 2

Reviewer 2 Report

The revised manuscript looks good. I suggest to publish the paper.